# Precision Medicine Study of Post-Exertional Malaise Epigenetic Changes in Myalgic Encephalomyelitis/Chronic Fatigue Patients During Exercise

**DOI:** 10.3390/ijms26178563

**Published:** 2025-09-03

**Authors:** Sayan Sharma, Lynette D. Hodges, Katie Peppercorn, Jemma Davis, Christina D. Edgar, Euan J. Rodger, Aniruddha Chatterjee, Warren P. Tate

**Affiliations:** 1Department of Pathology and Molecular Medicine, Dunedin School of Medicine, University of Otago, Dunedin 9016, New Zealand; shasa470@student.otago.ac.nz (S.S.); euan.rodger@otago.ac.nz (E.J.R.); 2School of Sport, Exercise and Nutrition, College of Health, Massey University, Palmerston North 4410, New Zealand; l.d.hodges@massey.ac.nz; 3Department of Biochemistry, School of Biomedical Sciences, University of Otago, Dunedin 9016, New Zealand; katie.peppercorn@otago.ac.nz (K.P.); ellje786@student.otago.ac.nz (J.D.); tinaedgar24@gmail.com (C.D.E.)

**Keywords:** ME/CFS, CPET, epigenetics, DNA methylation, post-exertional malaise

## Abstract

Post-exertional malaise (PEM) is a defining symptom of Myalgic Encephalomyelitis/Chronic Fatigue Syndrome (ME/CFS), yet its molecular underpinnings remain elusive. This study investigated the temporal–longitudinal DNA methylation changes associated with PEM using a structured two-day maximum repeated effort cardiopulmonary exercise testing (CPET) protocol involving pre- and two post-exercise blood samplings from five ME/CFS patients. Cardiopulmonary measurements revealed complex heterogeneous profiles among the patients compared to typical healthy controls, and VO_2_ peak indicated all patients had poor normative fitness. The switch to anaerobic metabolism occurred at a lower workload in some patients on Day Two of the test. Reduced Representation Bisulphite Sequencing followed by analysis with Differential Methylation Analysis Package-version 2 (DMAP2) identified differentially methylated fragments (DMFs) present in the DNA genomes of all five ME/CFS patients through the exercise test compared with ‘before exercise’. With further filtering for >10% methylation differences, there were early DMFs (0–24 h after first exercise test) and late DMFs between (24–48 h after the second exercise test), as well as DMFs that changed gradually (between 0 and 48 h). Of these, 98% were ME/CFS-specific, compared with the two healthy controls accompanying the longitudinal study. Principal component analysis illustrated the three distinct clusters at the 0 h, 24 h, and 48 h timepoints, but with heterogeneity among the patients within the clusters, highlighting dynamic methylation responses to exertion in individual patients. There were 24 ME/CFS-specific DMFs at gene promoter fragments that revealed distinct patterns of temporal methylation across the timepoints. Functional enrichment of ME-specific DMFs revealed pathways involved in endothelial function, morphogenesis, inflammation, and immune regulation. These findings uncovered temporally dynamic epigenetic changes in stress/immune functions in ME/CFS during PEM and suggest molecular signatures with potential for diagnosis and of mechanistic significance.

## 1. Introduction

Myalgic Encephalomyelitis/Chronic Fatigue Syndrome (ME/CFS) is a chronic, debilitating post-viral/stressor condition with complex pathophysiology, affecting many millions of people globally, that, for >95% of people affected, is lifelong, so new cases steadily add to the disease burden [1]. Recent understanding of genetic linkage to ME/CFS suggests there are clusters of multiple genetic variations encompassing 199 single-nucleotide polymorphisms that confer the susceptibility to develop the condition [2], after a major stress event like an infection with a virus or other microorganism, exposure to toxic chemicals, major surgery, or a significant stressful event in life. The ongoing ME/CFS is characterised by frequent relapses of an already poor state of health that reflects a sensitivity of those affected to small life stresses that are managed, without effect in healthy people, by the stress centre of the brain, a cluster of neurons in the paraventricular nucleus of the hypothalamus [3]. An unpublished new study publicly reported by Younger provided strong support for widespread inflammation both bilaterally and unilaterally across the brain of ME/CFS patients that could explain their neurological symptoms [4]. It supported a much earlier isolated PET study showing neuroinflammation [5]. The poor response to stress in ME/CFS, whether it be physical, emotional, or psychological, leads to the core symptom of ME/CFS, named post-exertional malaise (PEM), as one of a small set of defining symptoms that make up the clinical case definitions [6,7,8]. These are accompanied, however, by a much larger group of >200 reported symptoms, only some of which are experienced by each ME/CFS patient, reflecting their individual physiological responses [9,10]. The PEM symptom has been defined as “The inability to recover normally following physical, cognitive, or emotional exertion, [resulting in] a level of fatigue that is more profound, more devastating and longer lasting than is observed in patients with other fatiguing disorders” [8]. It encompasses any symptoms that occur after exertion, but it may be difficult for the patient to identify the exact cause. Even the smallest of daily tasks can be sufficient to invoke the response [11]. Core symptoms of ME/CFS that become worse in PEM include fatigue, cognitive functions, general pain and sleep disturbance, as well as flu-like symptoms and/or tender lymph nodes [12]. Onset can be immediate or delayed, and the relapse from PEM can last from days to months [12].

Cardiopulmonary exercise testing (CPET) is an important tool for evaluating ME/CFS [13] and can be used to induce PEM in research settings. Following this protocol, ME/CFS patients averaged two weeks more to recover from CPET, compared to sedentary controls, with 10% taking greater than three weeks [14]. A CPET study showed ME/CFS patients reached their anaerobic threshold well below that of the controls and decreased further in the second-day exercise test. Nevertheless, their ability to produce energy was similar to the controls [15]. A recent mixed-methods study for assessing PEM in ME/CFS patients concluded semi-structured qualitative interviews were superior to a visual analogue scale (VAS) assessment of symptoms to quantify disease severity over time, following a cardiopulmonary exercise test (CPET). The authors concluded combining qualitative interviews and quantitative measurements was a more comprehensive and effective way to assess PEM [16]. More precise quantitative methods are needed to assess the physiological changes that occur as PEM is developing in patients. The epigenetic study described here has that goal.

We have previously examined changes in the epigenetic DNA methylome before, during and beyond a relapse that lasted one and two months in two ME/CFS patients, respectively, and remarkably identified individual methylation sites on their DNA genomes that became either more hypo- or hypermethylated during the relapse but returned to their original levels when the relapse was over [17]. Major physiological categories identified during the relapse common to both patients included regulation of the immune system, energy production, metabolism, gene transcription, and long noncoding RNA (lncRNA) [17]. This suggested the DNA methylome was a sensitive indicator of changes in the expression of genes that were occurring during ME/CFS relapses. Our earlier epigenetic study of ME/CFS, the first to use Reduced Representative Bisulphite Sequencing (RRBS), identified many changes at methylation sites across the genome, some that could be linked to the expression of genes, particularly for regulation of the immune system, gene expression, and energy production [18]. Our most recent study that compared Long COVID and ME/CFS identified 12 gene promoter sites and six gene exons in common between the disease cohorts that were differentially methylated compared with healthy controls [19]. A clinical trial is currently underway to assess ‘genetic and epigenetic mechanisms’ of three major candidates involved in the function of the central nervous system–brain-derived neurotrophic factor (BDNF), catechol-O-methyltransferase (COMT), and histone deacetylases (HDAC) [20]. Epigenetic regulation of BDNF has been linked with symptoms and widespread pain in ME/CFS and fibromyalgia patients [21], and regulation of COMT linked with inflammation [22]. Other studies have used Illumina array technology to catalogue differentially methylated sites in the genomes of ME/CFS patients [23,24,25,26,27,28,29].

In this current study, we have carried out the cardiopulmonary exercise testing (CPET) protocol with two testing sessions to induce and study PEM from the perspective of how the DNA methylome changes in ME/CFS during the exercise. The detailed systematic study used precision medicine to collect data individually on five patients and, additionally, two healthy controls with diverse fitness levels. Significant ME/CFS-specific changes at 205 methylation sites in the DNA methylome were identified in response to the controlled exercise protocol.

## 2. Results

### 2.1. Two-Day Maximum Repeated Effort CardioPulmonary Exercise Testing (CPET)

The study was structured to characterise rigorously the molecular and clinical sequelae of post-exertional malaise (PEM) in participants over three days, with a focus on the timing of blood sample collection after exercise sessions. As shown in the schematic diagram in Figure 1, following a pre-exercise baseline assessment and an initial collection of blood samples (0 h), participants undertook a supervised exercise challenge on Day One. Twenty-four hours after the initial exercise test, a second blood sample (24 h) was collected immediately before a repeat exercise test on Day Two, enabling the assessment of physiological and molecular responses. A final blood sample was obtained on Day Three, 24 h after the second exercise test (48 h), facilitating the evaluation of prolonged or sustained effects of exertion. This design allowed for the systematic collection of biological samples and clinical data at critical timepoints, specifically aligned with exercise-induced stress and its aftermath. The protocol provides a robust framework for investigating the mechanisms underlying PEM in post-viral/stressor conditions like ME/CFS [12].

#### Summary of Cardiopulmonary Parameters for the Five ME/CFS Patients

Several important parameters examined on each of the two days of exercise were heart rate, maximum workload achieved, and VO_2_ peak—a measure of the greatest efficiency of the lungs, heart, and muscles to deliver and use oxygen. A key parameter for utilising energy, the anaerobic threshold, was derived. This is the point of the intensity of the exercise when there is a switch from aerobic to anaerobic metabolism. The longer it takes for this crossover is a measure of better “fitness”. These measurements for the five patients and two controls are described in Table 1 for the two days of the exercise protocol.

Three of the ME/CFS patients (ME007, ME0026, and ME0028) could not achieve the same peak workload on Day Two compared with Day One of the exercise protocol, while the other two (ME016 and ME024) maintained the same workload on both days. Two patients, ME007 and ME016, reached their anaerobic threshold at a lower workload on Day Two, and their VO_2_ peak fitness measure was also lower, whereas the other three (ME024, ME026, ME028) reached their anaerobic threshold at the same or greater workload. The VO_2_ peak of two of these patients (ME024, ME026) was higher, while ME028 was lower.

In ME016, the lower anaerobic threshold on Day Two was matched by a lower heart rate, whereas ME007 had a higher heart rate with a lower anaerobic threshold. ME024 and ME026 also had higher heart rates on Day Two while maintaining their anaerobic threshold, indicating the cardiovascular system was working harder to achieve it. While ME028’s heart rate on Day Two dropped along with maximum workload and with VO_2_ peak, her anaerobic threshold increased. Hence, there was a significant heterogeneity among the different cardiopulmonary parameters of the five ME/CFS patients in response to the exercise sessions, even though they were of similar age, the same sex, their symptom profiles were very similar, and their activity levels were equivalent.

Two controls were included in the study. One, despite recording a poor fitness level (VO_2_ peak), had consistent parameters on both Day One and Day Two, while the other had a VO_2_ peak within the 90th percentile, displaying her known superior athletic ability.

### 2.2. Overview of Differential Methylation Analysis of Post-Exertional Malaise

The workflow for identifying differentially methylated fragments (DMFs) in the PBMC DNA isolated from the blood samples taken during CPET in the context of post-exertional malaise (PEM) is shown in Figure 2. Methylated fragments contain clusters of CpGs each of which has the potential to be methylated at the C residue. If there is a statistically significant change in methylation through the timepoints at some or all of the CpGs in a fragment, then that fragment is referred to as being differentially methylated, or a differentially methylated fragment (DMF). Analysis was conducted with datasets from the five ME/CFS patients using the DMAP2 pipeline [30], with sequential filtering based on both statistical and biological thresholds across the three timepoints.

As shown in Figure 2 for the ME/CFS group, the bisulphite sequencing data had 188,343 methylated fragments that were present in all five ME/CFS patients at each of the three timepoints (0 h, 24 h, and 48 h). Application of ANOVA (*p* < 0.05) identified that 5272 of these were significantly differentially methylated through the timepoints. These fragments in the ME/CFS cohort were subjected to an additional biological relevance filter, requiring a greater than 10% methylation difference in any of the three pairwise timepoint comparisons: early response (first 24 h), late response (second 24 h), or gradual response (throughout the 48 h). After combining them from each timepoint, duplicates of fragments that were already recorded at an earlier timepoint were removed (Appendix A). Any DMFs found in this filtered ME/CFS dataset that were present in the two controls were also removed to yield a unique DMFs dataset in the ME/CFS patients arising during the two-day CPET. This stepwise approach enabled robust identification of ME/CFS-specific epigenetic changes associated with exertional stress from the CPET protocol, an indicator of PEM.

### 2.3. Principal Component Analysis of Differentially Methylated Fragments

To illustrate the global structure and temporal shifts in DNA methylation of the DMFs revealed by the ANOVA, principal component analysis (PCA) was performed on differentially methylated fragments (DMFs) identified across the three experimental timepoints (0 h, 24 h, 48 h) (Figure 3A). For the ME/CFS patients, PCA was conducted using the 5272 fragments found to show statistically significant differential methylation (ANOVA *p* < 0.05). This analysis revealed as expected distinct clustering of samples by timepoint, with a combined variance of 49.2% in PC1 and PC2. ME028 and ME024 showed some separation from the other three ME/CFS participants before the exercise protocol. At the 24 h and 48 h timepoints, four of the patients clustered tightly, including ME024, whereas ME028 was more of an outlier. This suggested dynamic methylation responses to exertion in all ME/CFS patients, with some heterogeneity in their “before exercise” methylation profiles, but more similarity in their responses to the exercise, with tighter clustering (Figure 3A). A PCA from the dataset of the two controls (Figure 3B) using the 14,054 statistically significant DMFs in their dataset (ANOVA *p* < 0.05) had a combined variance of 92.1% in PC1 and PC2. This was in contrast to the ME/CFS dataset where only 49.2% of the variability by contrast was captured in those two dimensions. A higher percentage of variance in the controls indicated these two components captured most of the total variability in the dataset in a two-dimensional space with most of its key information. Despite the similar temporal change profiles, in ME/CFS and controls, only 211 DMFs of the control’s 14,054 DMFs were in common with the 5272 DMFs identified in the ME/CFS patients. Although the two controls had widely diverse fitness levels, the PCA showed their DMF profiles were superimposed before exercise, and then closely overlapped still in their responses to the exercise. This was reinforced by the high percentages of data captured in the PC1 and PC2 components of the plot as described (Figure 3B).

### 2.4. Heatmap Analysis of Differentially Methylated Fragments in Controls and ME/CFS at Each Timepoint of CPET

To visualise temporal and disease-specific DNA methylation changes at individual sites during the CPET exercise protocol, heatmaps were generated using differentially methylated fragments (DMFs) further filtered. (Figure 3C,D). The DMFs, first identified as statistically significant (*p* < 0.05—Figure 2) were now further filtered (as shown in the workflow scheme in Figure 2) to include only those fragments showing a methylation difference greater than 10% in any one of the three possible pairwise timepoint comparisons. This resulted in 63 DMFs for early response (0 h–24 h), 44 DMFs for late response (24 h–48 h), and 120 DMFs for gradual response (0 h–48 h). The sets of DMFs passing this threshold in each comparison were then combined, resulting in a subfraction (209) showing >10% change in methylation. In the case of the two controls, although more of the significant DMFs had >10% change in methylation, only 4 of the 209 DMFs in the ME/CFS DMFs set overlapped. These 4 were removed so only the 205 ME/CFS-specific DMFs were further analysed. It should be noted that with five patients in the ME/CFS group but only two adjunct controls, this might have resulted in the higher number of DMFs (14,054) recorded for the controls.

The heatmap of ME/CFS-specific DMFs (Figure 3C) demonstrated heterogeneity (shades of colours) across the five patients at individual methylation fragments. The most hypermethylated block (0.6–1.0, colours orange to yellow) illustrates best the more dominant increase in hypermethylation (from 0 h to 24 h to 48 h) with the lighter yellow (1.0) more predominant at 48 h, but also the reverse in some fragments, which is hypomethylation. Dynamic changes in methylation patterns across the timepoints reflect the variable epigenetic responses from exertional stress observed among ME/CFS. By contrast, the two controls (Figure 3D) did not show the same heterogeneity before exercise and the changes at the 24 h and 48 h timepoints were also very similar, almost as though they were technical replicates from a single participant.

### 2.5. Methylation Dynamics in ME/CFS-Specific Promoter Fragments at Each Timepoint of the CPET Protocol

Among the 205 unique ME/CFS DMFs across the three timepoints of CPET, there were 24 gene promoter fragments in each of the five ME/CFS patients that could be linked to specific genes (Table 2). The promoters were defined as −5 kb to +1 kb from the transcriptional start site (TSS). The table shows the chromosome number and the start and end of each fragment, the number of CpGs within the fragment, the significance of the differential methylation change, and mean methylation % values at the three timepoints. The gene identity and its function are indicated.

There were several patterns of methylation change over the time course of the exercise of these identified promoters. Examples * of each type are shown in Figure 4.

(a)Continuous hypermethylation (*ERF **, *FAM3A*, *RNPS1*, *ZNF135*, *ZN785*, *OR2V1*),(b)Early hypermethylation (*DPM2 **, *ACR*, *APEX2*, *PRGGR3*, *ZAP70*, *ZDHHC9*),(c)Transient hypermethylation (*C20orf151 **, *C7*, *NPBWR2*, *FFAR2*, *ZG16B*),(d)Late hypermethylation (*TMEM187 **, *CPEB*),(e)Continuous decrease in hypomethylation (*MSR1 **),(f)Early hypomethylation (*CHST7 **),(g)Transient hypomethylation (*APCDD1L **, *FAM123B*),(h)Late hypomethylation (*DDX26B*).

An example of each pattern with individual patient data shown (coloured points) is illustrated in Figure 4. While there is some variation among the five patients in the degree of methylation change, in most cases, patients show the same trend.

### 2.6. Functional Enrichment of Genes Associated with Differentially Methylated Fragments

To investigate the biological relevance of the differentially methylated fragments (DMFs) identified in our PEM ME/CFS cohort, we performed gene ontology and pathway enrichment analysis using Metascape [31] on DMFs within promoters, gene exons and introns, that passed both statistical and biological thresholds (ANOVA *p* < 0.05, methylation difference > 10% in at least one timepoint comparison). Unique gene lists were derived after removal of overlapping fragments.

For the five ME/CFS patients, there were 113 unique genes (Appendix A) associated with the promoter and gene body DMFs in ME/CFS. Of these, the expression of 83 genes is influenced by hypermethylation of the DMFs in either of the early, gradual, or late phases (Appendix A), whereas the expression of 37 genes is influenced by hypomethylation (Appendix A). It should be noted that where gene-linked DMFs have shown hypomethylation or hypermethylation in the early phase and the opposite in the late phase (some members of the gradual set) we have included them in both methylation categories.

A Metascape analysis (Figure 5A) of those genes linked to hypermethylated DMFs showed enrichment notably in RHO GTPase cycle, JNK Cascade, DNA-templated Transcription, Wnt Signalling Pathway, Response to Steroid hormone, immune response, and Tyrosine Kinase Pathways. From these identified pathways it can be hypothesised that hypermethylation of the genes are likely regulating the inactivation or epigenetic shutdown of (i) immune regulatory signals, (ii) JNK and Steroid Pathways, and (iii) Transcriptional Processes from which patients fail to adapt after exertion, and experience immune/inflammatory dysfunction and fatigue.

A Metascape analysis (Figure 5B) of the genes linked to hypomethylated DMFs showed enrichment in cranial nerve development, endothelium development, Class A1 Rhodopsin-like receptors, and VEGFR2 signalling. These pathways suggest activation of angiogenesis and endothelium development, GPCR (hormonal and neurotransmitter signalling), and cranial nerve development, which may reflect abnormal neuro-sensory signalling. It is to be noted that the maximum heart rate of 4/5 ME/CFS patients decreased during the second exercise session as mentioned in Table 1, and it could be hypothesised that activation of the angiogenesis and endothelium development is an attempt to compensate for decreased cardiovascular activity in the body as seen in bpm and VO_2_ peak CP parameters.

The two healthy controls had 744 unique genes linked to the combined hypermethylated and hypomethylated DMFS (Appendix A) associated with promoters, and gene exons and introns. Functional relevance analysis collectively revealed significant enrichment for different processes such as neuron projection development, modulation of chemical synaptic transmission, hemopoiesis, muscle structure development, enzyme-linked receptor protein signalling pathway, skeletal system development, and cell adhesion (Appendix A). These findings underscore the broad engagement of genes involved in neurodevelopment, immune modulation, and cell communication in healthy subjects during the post-exertional period. In the healthy controls, the subgroup of DMFs associated with promoters (Appendix A) was associated with diverse functions enriched for axon development, response to growth hormone, pancreas development, signalling pathways, leukocyte migration, and synaptic activity (Appendix A). In stark contrast the subgroup of DMFs associated with promoters in ME/CFS patients were restricted only to positive activation of an immune response and an inflammatory response (Appendix A).

## 3. Discussion

This study was a precision medicine approach to determine how the cardiopulmonary parameters of individual ME/CFS patients compared in a CPET protocol before the exercise, during a maximum-effort exercise session, and then 24 h later during a second maximum-effort session. This was coupled to an epigenetic DNA methylome analysis of each patient before the exercise, 24 h after the first session, and 24 h after the second session when PEM was assumed to be developing. While this was a longitudinal study of the individual ME/CFS patients’ responses, two controls of diverse fitness were included to confirm that they produced the responses typically expected for age/sex-matched healthy individuals in CPET. Indeed, both controls showed similar parameters during both exercise sessions, although the actual values of each reflected the fitness levels of the two participants. For each individual, there were similar heart rates, similar max. workload, and similar VO_2_ peak on both days, with no decrease in anaerobic threshold (the workload points at which there is a switch to anaerobic metabolism to provide the needed energy). Indeed, the superior athlete control increased her VO_2_ peak and anaerobic threshold on the second exercise, suggesting she was able to work harder.

Each of the age/sex-linked group of ME/CFS patients showed individually specific cardiopulmonary responses during the CPET, as shown in Table 1. Four of the five had a lower maximum heart rate on the second exercise, and three of those four also were not able to achieve the same maximum workload; three had a lower VO_2_ peak on the second day, and the switch to anaerobic metabolism was at a lower workload in two participants. Hence, each patient showed some lower cardiopulmonary functions at the second exercise session, but not in all CPET parameters, and these varied among all five subjects. The results from the ME/CFS patients indicate a change in cardiovascular function consistent with the onset of PEM, in contrast to those of the two controls who performed as well on the second day.

A PCA of the identified DMF fragments identified showed well-separated clustering at each stage of the CPET protocol: before exercise, and 24 h after each of two exercise sessions (Figure 3A). There was scattering on the profile of the patients, indicating some variation in the DMFs among them, particularly before the exercise, despite the patients all being age/sex-matched. They were more tightly clustered in the changes in response to the exercise. This overview showed there were clear changes occurring in methylation at each of the timepoints in the ME/CFS patients in response to the exertion, over and above the changed epigenetic pattern from their ME/CFS condition present before exercise. In contrast, the two healthy control subjects (Figure 3B), completely overlapped in their epigenetic methylation status before the exercise, and still almost completely at each timepoint assessed during the exercise. The three timepoints were also widely separated.

The heatmap (Figure 3C) showed there were similar timepoint patterns among the five ME/CFS patients at each timepoint, but there was also significant variation at each fragment in the degree of methylation patient to patient. By contrast, the two healthy controls (Figure 3D) showed a different overall pattern of methylation with little variation between each of the two controls, while still exhibiting variation in the methylation pattern at the different timepoints.

Of the 205 ME/CFS-specific DMFs, 24 were associated with gene promoters and therefore could be linked to specific genes. The identities of these genes are shown in Table 2. These were predominantly concerned with immune functions, signal transduction, and gene expression (transcription factors/RNA processing). Nineteen of the promoter fragments showed hypermethylation and five showed hypomethylation, suggesting both decreases and increases in gene expression were likely occurring. There were eight different patterns of methylation change at these promoters over the three timepoints. At the hypermethylation regions, six promoters had early hypermethylation from the first exercise that became stronger after the second, six promoters had early hypermethylation but with no further change after the second exercise, five promoters had early hypermethylation that reversed after the second exercise, and two promoters had late hypermethylation after the second exercise. At the promoter regions where hypomethylation was occurring, one had gradual increasing hypomethylation, two promoters had early hypomethylation that reversed at the second exercise, one promoter had early hypomethylation with no further change, and one promoter was hypomethylated late.

From the Metascape analysis of the 113 genes identified (Appendix A) that could be reliably linked with changing methylation levels in promoters and gene bodies as a result of exercise, eight—ZDBF2, MSR1, CHD7, RXRA, ACSBG1, CACNA1H, MSI2, KLHL26—had also been identified as being differentially methylated compared with healthy controls in a non-exertion single timepoint study [19]. This implies there was further change in methylation as a result of the exercise stress. ZDBF2 paternal allele is expressed in lymphocytes and the hypothalamus and contains DBF4-type zinc finger domains [32]. MSR1 encodes the macrophage scavenger receptor protein, primarily found on the surface of macrophages and involved in innate and adaptive immunity, with a role in lipid metabolism [33]. The gene product of CHD7 binds to a helicase and has an important role in gene regulation through chromatin remodelling, targeting active enhancer regions [34]. RXRA is a ligand-activated transcription factor associated with many functions as an obligatory heterodimerisation partner for a range of nuclear hormone receptors for signalling through the pathways mediated by those receptors. It is indispensable in adult hematopoiesis [35]. ACSBG1 is associated with acyl-CoA synthesis and involved in fatty acid metabolism, but importantly, it is also involved in the differentiation and function of T cells, particularly TH17 and regulatory T cells. It is implicated in inflammation [36]. CACNA1H encodes a subunit of a calcium ion channel with an important role in regulating neural excitability. Effects of mutations in the gene indicate a role in the immune and gastrointestinal systems [37] This gene was also identified in a dataset of an earlier study of ME/CFS patients connected to a differentially methylated site, which suggests modification of this calcium channel may play a significant role in ME/CFS and the core PEM symptom [18,19]. MSI2 encodes an RNA-binding protein that regulates gene expression post-transcriptionally with an important role in stem cell maintenance, cell proliferation, and differentiation [38]. KLHL26 may play a role in endoplasmic reticulum (ER) mitochondrial signalling [39].

Sankey plots were produced to compare the major physiological molecular categories affected by the genes linked to the differentially methylated promoters and gene body genomic regions at an early response (28 genes after the first exercise) (Figure 6A), at a late response (53 genes after the second exercise) (Figure 6B), and with a gradual response (67 genes) (Figure 6C). At the early response with fewest genes, the categories linked to the differential expression included signal transduction, lipid metabolism, neural development, protein modification, and metabolism. The later response at 48 h had significantly more genes, some of which were also linked to signal transduction, but additionally different categories cytoskeleton, transcription, immune system, and ion transport. The categories linked to the continuing gradual methylation changes also included signal transduction, transcription, and the immune system, but together with new categories of RNA processing and of structural proteins. These results suggest that changes were occurring early by 24 h after one exercise session, but they became more pronounced later. The genes and their characteristics are described in Appendix A for the early, late, and gradual response-linked genes.

Five genes, *ACOT9*, *DUSP9*, *LSM14A*, *TOP1MT*, *SEMA6B*, linked to DMFs in response to exercise stress in the current study were also found in a longitudinal study through a stress-related relapse of two of the same patients [17]; four in one patient and the fifth in the second patient. Despite this diversity, the physiological/biochemical consequences of the relapse were very similar in the two patients. *ACOT9* is a mitochondrial acyl-CoA thioesterase of unknown function [40], *DUSP9* is phosphatase protein associated with insulin signalling and dephosphorylates MAPK/JNK proteins [41], *LSM14A* encodes a protein important in innate immunity and a key component of P-bodies involved in mRNA degradation and translation regulation [42], *TOP1MT* is a mitochondrial DNA topoisomerase that plays a role in managing topological stress during mtDNA replication and transcription [43], and *SEMA6B* is a member of Semaphorin family and play a major role in neuronal development, synapse formation, and axon guidance [44]. It is difficult without further study to speculate what role changes in expression of these genes may have in the initial PEM response to stressful exertion, and an actual relapse equivalent to PEM, even if the initiating event was not clear. However, it is not surprising that there would be common changes in the epigenetic code between the two events.

## 4. Materials and Methods

### 4.1. The Cardiopulmonary Exercise Training (CPET) Protocol

Five age/sex-matched ME/CFS patients (Table 3) who had been diagnosed by a specialist ME/CFS Clinician, Dr. Rosamund Vallings in New Zealand, and two age/sex-matched healthy controls were added to the study. The ME/CFS cohort filled in detailed questionnaires seeking the origin and course of their illness and the symptoms associated with it. The symptoms were graded on a severity scale. All were significantly affected but with moderate functional activity and were able to go to a community pathology laboratory to give their blood samples. Although the causes attributed to the development of the illness of the ME/CFS patients were diverse, the clinical phenotype of these recruited patients was very similar to the core symptoms consistent with the Canadian Consensus Clinical definition expressed in their very similar lives as university students. They had relapses, required occasional rest days, managed their physical activity around campus carefully, showed post-exertional stress at exam times, and required extra time consistent with effects on their cognitive abilities. All participants undertook a two-day maximum-effort cardiopulmonary exercise training protocol (CPET). The study was approved by the Central Health and Disabilities Committee (13/CEN/203) and Southern health and Disabilities Committee (17/STH/188) and conducted in accordance with the Declaration of Helsinki. Written informed consent was obtained from all participants.

Both ME/CFS patients and controls were instructed to avoid strenuous exercise prior to partaking in the study for 24 h, caffeine for 4 h, and nicotine or food for 2 h. Following an initial 5 min rest period, blood pressure (BP) and heart rate (HR) were recorded, followed by the measurement of height and weight. Participants cycled on an electromagnetically braked cycle ergometer (Lode Excalibur Sport, Gronigen 9747, The Netherlands) at between 50 and 80 rpm. Cycle seat height was positioned to approximately 175 degrees of knee extension, and the same seat height was used on both occasions. Starting at 15 W, the intensity was increased at a rate of 15 W/min. The test was terminated voluntarily by each study member when they were unable to maintain a pedal frequency of 50 rpm, or the ACSM termination criteria were met [45]. Heart rate (Polar Electro, Kempele 90440, Finland) and RPE (Borg 6–20 Scale [46]) were collected in the last 15 s of each workload, and BP was measured every 3 min. Online respiratory measurements were measured using a two-way breathing valve, and recorded breath by breath. The online breath-by-breath system (Cosmed Sri, Rome 00041, Italy) was calibrated within 30 min before each test. Standard calibration of the metabolic cart was used. Ventilatory or anaerobic threshold was identified from expired gases using the V-slope method [47]. The laboratory environment remained at 19 degrees Celsius during all testing procedures. Blood pressure and heart rate were monitored during recovery until the participant was within 20 bpm and 10 mmHg of resting measures. Participants repeated the cardiopulmonary exercise protocol (CPET) 24 h later.

### 4.2. Peripheral Blood Mononuclear Cell (PBMC) Isolation

Patients completed a brief survey of their health status on the day at the time of each blood collection. Blood fractions were processed within 24 h. Peripheral blood mononuclear cells (PBMCs) were isolated from whole blood as described [48] by layering it onto Ficoll-Paque (Cytiva, Uppsala, Sweden), followed by centrifuging at 400× *g* to separate plasma from PBMCs and the PBMCs from other cells. The removed PBMC layer was diluted with PBS and then pelleted at 100× *g*. The pellet was washed in PBS and stored at −80 °C in a solution of foetal calf serum containing 10% DMSO.

### 4.3. DNA Extraction

DNA was extracted from 200 μL of the PBMC fraction using the Illustra Blood Genomic Prep Mini Spin Kit (GE Healthcare UK Ltd., Chalfont St Giles, Bucks, UK) following the manufacturer’s instructions. Elution was performed using the kit’s EB buffer, and DNA concentration was measured with a Qubit 2.0 fluorometer according to the Qubit dsDNA HS Assay Kit protocol ThermoFisher Scientific, Waltham, MA, USA).

### 4.4. Reduced Representation Bisulphite Sequencing

Reduced Representation Bisulphite Sequencing (RRBS) libraries were prepared following previously established protocols [49,50]. We have utilised the RRBS platform successfully in ME/CFS and also in LC patients [17,18,19]. In summary, 500 ng of genomic DNA was digested using 160 U of the MspI restriction enzyme. After end repair and adenylation of the 3′ ends, adaptors were ligated to the DNA fragments. Bisulphite conversion was carried out using the EZ DNA Methylation Kit (Zymo Research Corp, Irvine, CA, USA) according to the manufacturer’s instructions. A semi-quantitative PCR was performed on the bisulphite-converted DNA to determine the optimal number of amplification cycles required for the final large-scale PCR of the complete library. Following PCR amplification, DNA was size-selected using magnetic beads (AMPure XP beads from the TruSeq DNA nanokit -San Diego Illumina, Inc., San Diego, CA, USA) as described in [48], isolating fragments between 40 and 220 bp to construct the RRBS libraries while minimising adaptor contamination. The purified DNA was assessed for quality using a BioAnalyzer (Agilent, Santa Clara, CA, USA) and Qubit (ThermoFisher Scientific, Waltham, MA, USA) measurements, followed by further purification with AMPure XP Beads (Beckman Coulter, Brea, CA, USA).

#### 4.4.1. DNA Sequencing

Samples were sequenced at the Otago Genomics and Bioinformatics Facility. After sequencing, raw FASTQ files were assessed for adaptor sequences and trimmed accordingly. The processed reads were then aligned to the human genome (GRCh37/hg19) using Bismark, generating BAM files for subsequent differential methylation analysis.

#### 4.4.2. Statistical Analyses

Analyses were conducted using the updated version of the DMAP pipeline [50,51], namely DMAP2 [30] on a macOS computer, to examine methylation changes across fragments ranging from 40 to 220 bp. DMAP2 employed an ANOVA F-test to compare patient and control groups, ensuring that only fragments with data available for all individuals in each group were included. A raw *p*-value threshold of <0.05 was applied. We have used a *p*-value threshold without false discovery rate correction in order not to lose true positives from this analysis, considering our sample size is low. Genomic features overlapping with these fragments were identified using the in-built Geneloc function of DMAP2. For the function of the genes, Gene Cards [52] were used, and to assign the functional categories, ChatGPT 4.5 was used to assign the functional categories based on Gene Cards’ functions.

## 5. Conclusions

To investigate the early stages of the development of post-exertional malaise, five ME/CFS patients undertook a two-day maximum-effort cardiopulmonary exercise testing protocol (CPET) together with a longitudinal molecular analysis of the changes in their epigenetic DNA methylomes in response to the exercise. The study showed the age/sex-matched ME/CFS patients, with similar overall clinical phenotype and functional activity, exhibited significant variation in their cardiovascular functions and variation in their epigenetic changes but that led to similar dysfunctional physiology and affected biochemical pathways. The ME/CFS patients showed a deterioration in their cardiopulmonary parameters on Day Two, in contrast to the controls, and suggested energy production and not cardiovascular function was the limiting factor in some cases, but cardiovascular function in others. The DNA methylome changes in the ME/CFS patients after each day of the exercise were almost all ME/CFS-specific. All patients and a control had normative poor fitness [53], but unlike the ME/CFS patients that control still performed as well in the CPET on successive exercise days. Functional enrichment revealed affected pathways of endothelial function, morphogenesis, inflammation, and immune regulation. These findings uncovered temporally dynamic epigenetic alterations in stress/immune functions in ME/CFS during PEM.

## 6. Future Directions

The affected genes identified in this study are generally linked to multiple and diverse complex aspects of human physiology in multiple tissues, and that creates a challenge for finding useful, specific biomarkers suitable as targets for the treatment of dysfunctions like immune dysfunction or inflammation. Teasing out promising candidate targets will need to be integrated first with patient RNA and protein multiomic studies. Cell-free DNA found in plasma reflecting all tissues might be more informative for exploring biomarkers derived from the DNA methylome molecular signatures than those derived from the DNA of peripheral blood mononuclear cells (PBMCs), despite the key dysfunction of the immune system in ME/CFS.

## Figures and Tables

**Figure 1 ijms-26-08563-f001:**
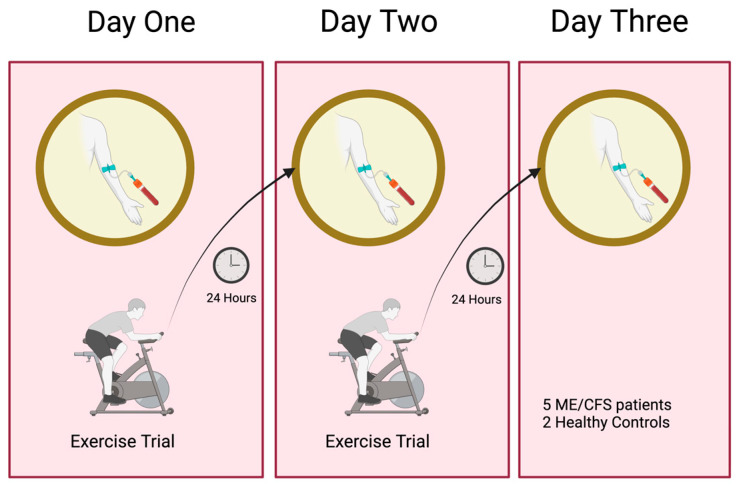
Schematic overview of the post-exertional malaise (PEM) Study Protocol. Participants underwent a pre-exercise baseline assessment, and a blood sample (0 h) was taken before the CPET protocol. It was followed by two consecutive daily exercise trials (Day One and Day Two) and follow-up assessments with blood sampling 24 h after each exercise session on Day Two and Day Three, enabling the temporal mapping of physiological and molecular responses associated with PEM. There were five ME/CFS participants and two controls, and DNA methylomes were analysed at the three timepoints for each participant.

**Figure 2 ijms-26-08563-f002:**
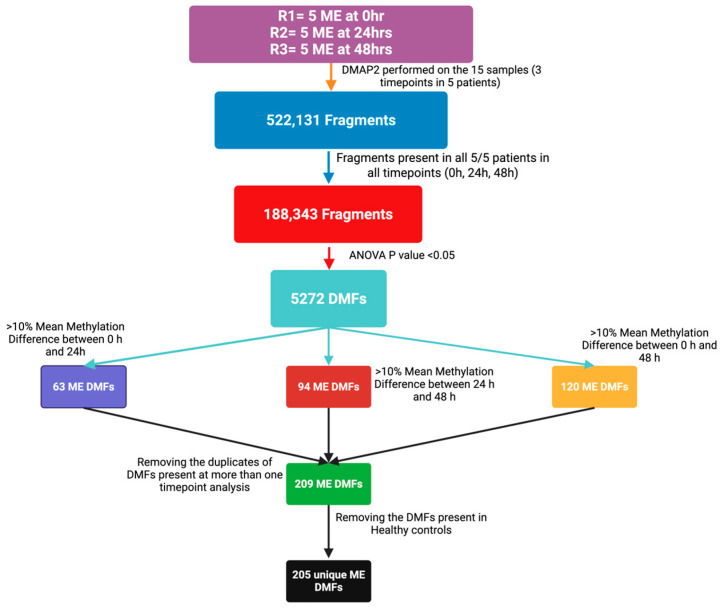
Workflow for the identification and classification of differentially methylated fragments (DMFs) in control and ME/CFS samples. Bisulphite sequencing data from ME/CFS patients, sampled at 0 h, 24 h, and 48 h, were processed using DMAP2 [30]. Only methylated fragments (188,342) present in all samples across the timepoints were further analysed. DMFs were first identified by ANOVA (*p* < 0.05). The individual DMFs were subsequently filtered to retain only those with a methylation difference greater than 10% in any of the three pairwise timepoint comparisons: 0 h vs. 24 h (early response), 24 h vs. 48 h (late response), or 0 h vs. 48 h (gradual response). Duplicates of fragments (already recorded in one category) were removed, together with the only 4 of the 209 fragments that were not ME/CFS-specific as they were also in the two controls. The resulted in 205 unique ME/CFS-specific DMFs for downstream analysis. While the workflow shown is only representative of that for the ME/CFS patients, a similar workflow was carried out for the two controls.

**Figure 3 ijms-26-08563-f003:**
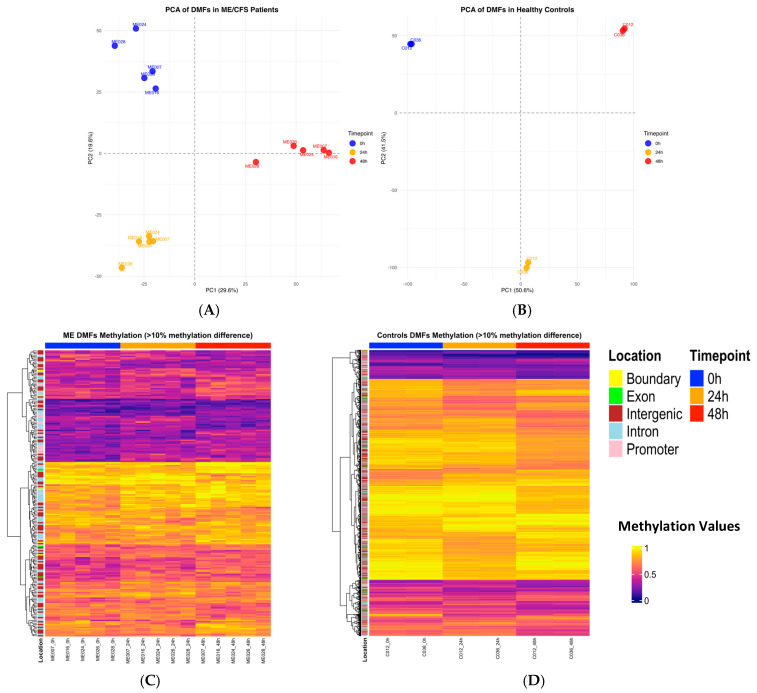
Analysis of differentially methylated DNA fragments across the CPET exercise protocol for the ME/CFS patients. (**A**) PCA of 5272 DMFs identified in ME/CFS patients (*p* < 0.05), showing distinct separations between before exercise (0 h), after the first exercise session (24 h), and after the second exercise session (48 h) indicating dynamic methylation shifts following exertion in ME/CFS (0 h—blue, 24 h—yellow, 48 h—red). (**B**) PCA of 14,054 DMFs identified in healthy controls (*p* < 0.05), showing distinct and tight clustering of both samples at each timepoint (0 h—blue, 24 h—yellow, 48 h—red. (**C**,**D**) The ME/CFS-specific DMFs (**C**) and the control DMFs (**D**) were filtered for >10% (0.1) methylation difference in any pairwise timepoint comparison; 0 h vs. 24 h, 24 h vs. 48 h, or 0 h vs. 48 h) during onset of post-exertional malaise (PEM). The heatmaps show the three timepoints, illustrating heterogeneity among the DMFs of the five patients at each timepoint, but near homogeneity between the two controls. The annotation column bar at the top of each heatmap represents the timepoints of the CPET (0 h—blue, 24 h—orange, 48 h—red). The colour gradient (indigo to yellow) shown in the key indicates the degree of methylation of the fragments from hypomethylation (indigo) to hypermethylation (yellow). It indicates the colours matching specific methylation values. The locations of the DMFs are annotated beside the heatmaps, gene promoters (−5 kb to +1 kb from the transcriptional start site—TSS), exons, introns, and intergenic elements (>5 kb upstream from the nearest TSS), and intron–exon boundary elements.

**Figure 4 ijms-26-08563-f004:**
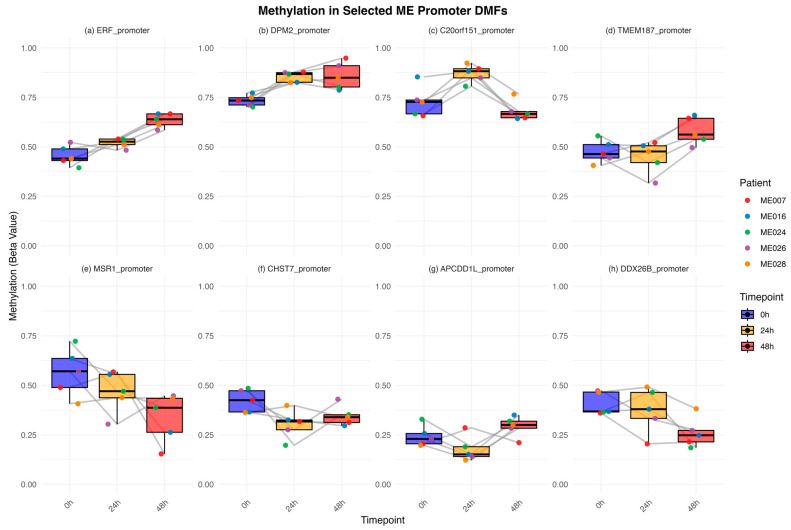
Dynamics of DNA methylation changes in examples of ME/CFS-specific promoter fragments for all five ME/CFS patients across timepoints of the CPET protocol. (**a**) *ERF*—continuous hypermethylation, (**b**) *DPM2*—early hypermethylation, (**c**) *C20orf151*—transient hypermethylation, (**d**) *TMEM187*—late hypermethylation, (**e**) *MSR1*—continuous hypomethylation (**f**) *CHST7*—early hypomethylation, (**g**) *APCDD1L*—transient hypomethylation, (**h**) *DDX26B*—late hypomethylation. Box plots with overlayed lines depict the distribution and trajectory of methylation at each promoter for each patient. The mean and first and third quartiles are shown, with the patients individually coloured (see key). Before exercise (0 h) is in blue, 24 h in yellow, and 48 h in red. Associated gene IDs are displayed at the top of each subplot.

**Figure 5 ijms-26-08563-f005:**
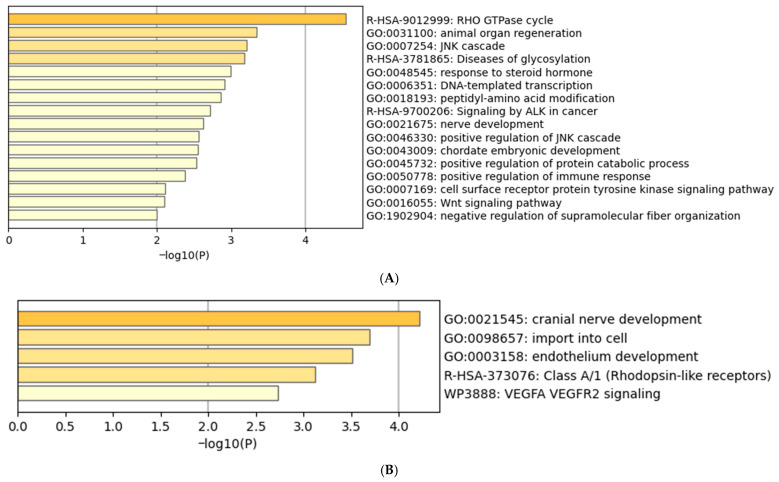
Functional pathway enrichment of genes associated with hyper-and hypomethylated DMFs in ME/CFS. (**A**) Top GO terms for unique ME/CFS hypermethylated gene-linked DMFs at promoters and gene bodies. (**B**) Top GO terms for unique ME/CFS hypomethylated gene-linked DMFs at promoter and gene bodies.

**Figure 6 ijms-26-08563-f006:**
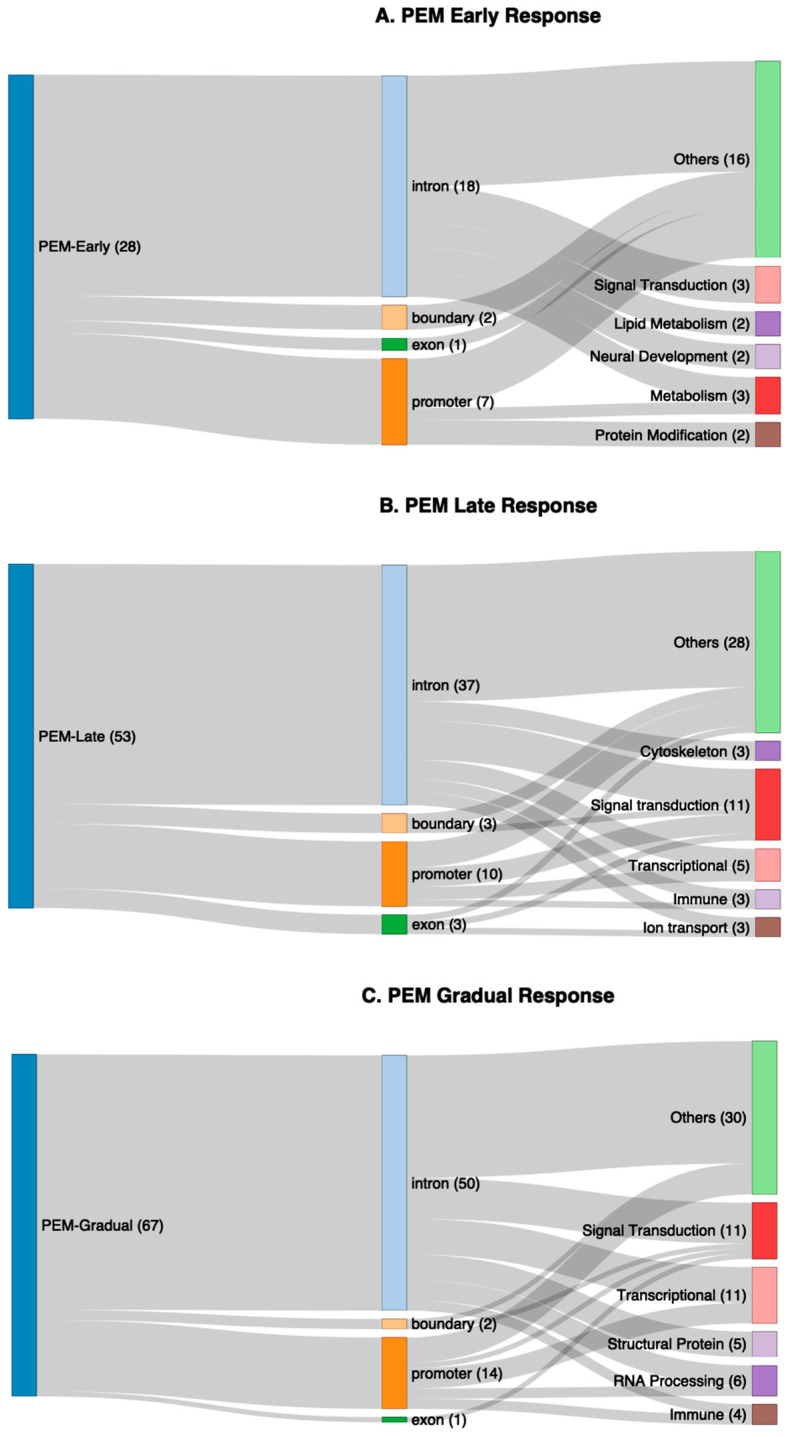
Sankey plots showing physiological/molecular categories associated with genes linked to differentially methylated regions in response to the CPET exercise protocol. (**A**) Genes (28) linked to differentially methylated promoters and gene bodies at early response as described in Appendix A. (**B**) Genes (53) linked to differentially methylated promoters and gene bodies (described in Appendix A) identified as a late response after the second exercise. (**C**) Genes (67) linked to differentially methylated promoters and gene bodies (described in Appendix A) identified gradually through the exercise. The locations of the sites within the genes (intron, exon, promoter, exon/intron boundary) and the main functional categories of molecular physiology affected, categorised by AI based on the functions of the genes.

**Table 1 ijms-26-08563-t001:** Cardiopulmonary profiles of ME/CFS patients and controls.

CPET Parameter	ME007	ME016	ME024	ME026	ME028	CO12	C036
**Day One** Maximum HR (bpm)	178	184	187	197	157	188	183
**Day Two** Maximum HR (bpm)	172 *	188	171 *	189 *	138 *	191	181
**Day One** Max.Workload (Watts)	135	195	135	210	180	285	165
**Day Two** Max.Workload (Watts)	120 ^#^	195	135	180 ^#^	165 ^#^	285	165
**Day One** VO_2_ peak (mL/kg/min)	25.4	34.4	29.1	30.1	35.1	49.4	21.8
**Day Two** VO_2_ peak (mL/kg/min)	26.1	33.2 ^++^	31.6	29.0 ^++^	31.1 ^++^	50.7	22.0
**Day One** RER	1.24	1.09	1.12	1.13	1.25	1.20	1.04
**Day Two** RER	1.13	1.11	1.19	1.04	1.30	1.17	1.17
**Day One** AT Workload (Watts)	60	105	75	135	90	195	75
**Day Two** AT Workload (Watts)	45 **	60 **	75	150 ^##^	120 ^##^	210 ^##^	75

Heart rate (HR) is in bpm, Workload is in Watts, VO_2_ peak in mL/kg/min, RER is the Respiratory Exchange ratio (VCO_2_/VO_2_), AT is anaerobic threshold, * indicates lower max. HR on Day Two, ^#^ indicates lower max. workload on Day Two, ^++^ indicates lower VO_2_ peak on Day Two, ** indicates where anaerobic threshold has occurred at lower workload on Day Two, and ^##^ where anaerobic threshold has occurred at a higher workload on Day Two.

**Table 2 ijms-26-08563-t002:** Gene promoters showing differential methylation in the ME/CFS patients during the exercise protocol.

Chr	Start	End	CpG	*p*-Value	Mean % Methylation (0 h)	Mean % Methylation (24 h)	Mean % Methylation (48 h)	GeneID	Functional Category
16	2880299	2880358	3	0.042	31	37	27	*ZG16B*	Cell Migration
X	55026095	55026179	3	0.014	26	35	38	*APEX2*	DNA Repair
5	40909532	40909656	1	0.009	28	38	20	*C7*	Immune
2	98329337	98329462	3	0.042	25	34	37	*ZAP70*	Immune
8	16425403	16425502	1	0.033	57	47	38	*MSR1*	Immune
9	130700685	130700757	6	0.001	74	85	85	*DPM2*	Metabolism
19	35939798	35939864	3	0.015	29	36	23	*FFAR2*	Metabolism
X	153743994	153744125	7	0.04	26	30	36	*FAM3A*	Metabolism
X	46433376	46433503	18	0.029	42	30	36	*CHST7*	Metabolism
X	128977778	128977938	12	0.003	26	36	40	*ZDHHC9*	Protein Modification
X	150863035	150863196	7	0.046	56	68	66	*PRRG3*	Protein Modification
22	51172813	51172946	4	0.014	61	73	70	*ACR*	Reproduction
X	134655182	134655341	14	0.046	40	36	28	*DDX26B*	RNA processing
16	2322947	2323056	4	0.046	57	64	69	*RNPS1*	RNA processing
5	180554560	180554676	1	0.003	84	88	96	*OR2V1*	Sensory Perception
20	62738880	62738959	4	0.004	16	23	12	*NPBWR2*	Signal Transduction
20	57090702	57090793	2	0.025	23	17	30	*APCDD1L*	Signal Transduction
X	63425502	63425652	11	0.041	31	29	42	*FAM123B*	Signal Transduction
19	42760491	42760639	3	0.00003	46	52	62	*ERF*	Transcription
20	48805866	48805958	1	0.014	79	75	92	*CEBPB*	Transcription
16	30597821	30597994	7	0.025	39	43	53	*ZNF785*	Transcription
19	58571275	58571401	5	0.027	30	34	41	*ZNF135*	Transcription
20	61005667	61005710	1	0.0008	74	86	69	*C20orf151*	Unknown
X	153233106	153233193	3	0.030	48	45	56	*TMEM187*	Unknown

Chr is the chromosome number, ‘start and end’ mark the genomic position of the fragment, CpGs refer to the number of sites within the fragment, *p* marks the significance of the DMF. The mean methylation percentages for the data of the five patients are indicated at each of the three timepoints. GeneID gives the gene identity, and functional category provides the broad function of the linked gene.

**Table 3 ijms-26-08563-t003:** Patient Characteristics including Age, Sex, Height, Weight, and BMI.

Characteristics	ME/CFS Mean +/− SD	Controls Mean +/− SD
Sex	5 females	2 females
Age (years)	22 ± 3.63	25 ± 0
Height (m)	1.70 ± 0.03	1.62 ± 0.57
Weight (kg)	62.97 ± 9.93	75.1 ± 14
BMI (kg·m^2^)	21.84 ± 3.87	28.86 ± 7.34

## Data Availability

All datasets generated and analysed during this current study are available in the GEO database accession number: GSE304805.

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
