# Peer review of "Precision Medicine Study of Post-Exertional Malaise Epigenetic Changes in Myalgic Encephalomyelitis/Chronic Fatigue Patients During Exercise"

_ijms, 2025, doi:10.3390/ijms26178563_

Round 1
Reviewer 1 Report
Comments and Suggestions for Authors
The manuscript "Post-Exertional Malaise epigenetic changes in Myalgic Encephalomyelitis/Chronic Fatigue patients during Exercise" by S. Sharma et al presents an interesting study of changes in the patterns of genomic DNA methylation in lymphocytes of Myalgic Encephalomyelitis/Chronic Fatigue (ME/CFS) patients undergoing Cardiopulmonary Exercise Testing (CPET). Authors conducted a very thorough analysis of DNA methylation data and presented interesting results.
However, there are some minor issues.
Only five ME/CFS patients and two controls participated in this study. Although this study was designed as precision medicine study, results were averaged, and statistics were applied for patients and controls. The sample size is very small for generalized conclusions. I think that authors should specify that this is a pilot study, and I think this should be reflected in the title.
It is impossible to correlate clinical symptoms severity in such small cohort. However, in future studies, other researchers will want to compare results and correlate results of molecular measurements with symptoms. It would be beneficial if authors present results of questionnaires in aggregated form, probably as an additional supplementary table.
A very minor suggestion – the abbreviation “DMAP2” in the abstract should be spelled out.
Author Response
Please see the the attachment in the box

Reviewer 2 Report
Comments and Suggestions for Authors
This paper aims to use the 2-day CPET protocol to induce PEM in ME/CFS patients and identify DNA methylation patterns that differ between healthy controls and subjects with ME/CFS at baseline and 24 hours after each exercise bout. Specific genes and pathways relevant to the pathophysiology of ME/CFS were identified.
Overall, the manuscript is excellent. The study used one control subject with poor fitness and one control subject with elite fitness level to demonstrate the homogeneity of a healthy response to exercise, in contrast with the heterogeneous response of the ME/CFS subjects. A balanced number of controls and subjects would have been ideal, however, the near-uniformity of DMF patterns between the two controls make this criticism inconsequential. This is quite striking, and I suggest that the authors consider moving the contents of Supplementary Figure 2 into the main Figure 2 as additional panels to increase the impact of Figure 2. There are several reference problems that must be addressed. Most of the requested revisions could be considered minor, but there are quite a few and revising a figure is no small undertaking, hence the "major revisions" classification.
Specific comments:
Line 47: Self-citation of a paper that did not generate primary data to support the claim is inappropriate. Please cite a paper that generated the data (frequency, recovery rate, etc).
Line 53-55: The authors state that the “stress centre of the brain” in ME/CFS patients is unable to manage normal stresses handled by healthy people, and cite their own hypothesis paper to support the statement. This is inappropriate. Please do not cite a hypothesis paper as fact, especially self-citation. This statement needs to be supported by evidence or removed.
Line 65: “Even the smallest of daily tasks is sufficient to invoke the response.” This statement should be qualified, e.g. replace “is” with “can be”.
Line 65: “…why the stress centre overreacts to even small stresses in ME/CFS patients.” The supporting reference for this claim is a self-cited hypothesis paper. It is enough to say that neuroinflammation has been documented.
Line 75-76: “Psychometric evaluation of a visual analogue scale (VAS) has been used for assessment of symptoms [14].” Reference 14 does not support the statement, although it is a relevant study whose findings should be included in the introduction (this may have been an editing error with a resultant misplaced citation).
Lines 76-79: “A recent study conducted semi-structured qualitative interviews at the same time points as VAS measures that quantify disease severity of PEM following a cardiopulmonary exercise test (CPET) and was able to successfully monitor changes in PEM over time [15].” I think I know what you mean, but the sentence is very confusing (this goes beyond grammar). Please rephrase. If VAS and QI are the only way to measure PEM, this should be stated clearly in this paragraph, since it strengthens the importance of your findings.
Line 148-163: The Results section should contain statements of fact without interpretation. Remove phrases that begin with “suggesting” or “suggested” (four phrases). Move interpretation of these results to the Discussion section.
Line 172 etc: Description of differential methylation should be included in the text (not just the figure), to clarify what is being compared (time points in the same set of patients, not between patients and healthy controls).
Line 188: The phrase “two exercise CPET” should be replaced with “2-day CPET” or something similar.
Line 206: The figure legend should make clear that the figure is a representative workflow for ME/CFS patients (only), and a similar workflow was used for controls.
Line 233-234: “the PCA showed their DMFs 233 profiles were superimposed before exercise, and then in their responses to the exercise 234 were also closely overlapping” Most readers will not look at the Supplementary Figures (unfortunately), and this is visually striking compared to ME/CFS data. Consider adding it as a panel to Figure 2 in the main text.
Lines 235-236: “This was reinforced by the high percentages of data reflected in the PC1 and PC2 components of the plot described in supplementary Figure S1.” First, many readers will not be familiar with PCA, and it might be worth explaining briefly, or rephrasing the statement for a reader without PCA knowledge. Consider adding control data as a panel to Figure 2 and pointing out the difference between ME/CFS and controls (axis %), as the differences are striking.
Lines 400-403: “There was evidence, however, from the results of a change in cardio-vascular function 400 consistent with the onset of PEM, in contrast to the two controls who performed equally 401 well or better on the second day, even though one control scored a relatively low fitness 402 level, like the ME/CFS patients.” This sentence is very confusing. Evidence of what? Please rephrase.
Line 404: “A PCA of the DMF fragments identified showed..” This is confusing. I think you mean “identified DMF fragments” but I’m not sure. Please rephrase to make your meaning clear.
Line 449-450: “It is indispensable in adult hematopoiesis.” A reference is required to support this statement. If the statement is supported by reference 34, add it to the previous sentence in some way.
Line 452-453: “It is implicated in inflammation.” See previous comment for Lines 449-450.
Lines 455-458: “This gene was also identified in a dataset of an earlier study of ME/CFS patients connected to a differentially methylated site, which suggests modification of this calcium channel may play a significant role in ME/CFS and the core PEM symptom.” This statement needs a supporting reference.
Line 467: “…the differentially expression…” I think you mean “differential expression”.
Lines 607-643: The Conclusion section needs to be rewritten. It should be 3-5 sentences stating what the results mean. It should not contain methods or results. The statements in the “Future Directions” section would probably suffice.
Comments on the Quality of English LanguageThere are some issues with clarity that may be related to English Language skill. These are detailed in the previous section. Overall the paper is well written.
Author Response
Pleas\e see the attachment in the box

Round 2
Reviewer 2 Report
Comments and Suggestions for Authors
Thank you for making the suggested changes. I am sorry about the back-and-forth regarding the Conclusions section-- I know how frustrating it can be when two reviewers ask for opposite things. Regarding the neuroinflammation citation, I will leave it up to the editor whether or not it is appropriate to cite a blog post. Dr Younger does not have a preprint imminent for you to cite.